# Techno-Economic Study of a Distributed Renewable Power System for a British Winery

Sophie Hall-Smith [1], Yaodong Wang [1,*] and Ye Huang [2]

1 Department of Engineering, University of Durham, Durham DH1 3LE, UK; sophhs913@gmail.com
2 Belfast School of Architecture and the Built Environment, Ulster University, Newtownabbey, Co., Antrim BT37 0QB, UK; y.huang@ulster.ac.uk
* Correspondence: yaodong.wang@durham.ac.uk

**Abstract:** This paper analyses and evaluates a design for a distributed renewable power system for a British winery. A winery in Wiltshire, England, is used for a case study. The consumption of this winery is first analysed, then potential means of generation are discussed. The resulting design is a combination of 156 $1.6 \times 1$ m$^2$ photovoltaic panels; a $2 \times 12$ m$^2$ modular anaerobic digester using winery and farm waste to produce 0.00287 kg/s of biogas; and a biogas combined heat and power generator to supply 188 MWhe and 44 MWht per year. This was analysed technically, using ECLIPSE, and economically. The design would reduce the carbon footprint of a winery by 41,100 kgCO$_2$/year. The techno-economic performance was compared with traditional power generation means; the designed system is technically viable, and financial incentives allow it to compete economically with alternatives. The cost of the design varies more with technology price than incentives, demonstrating that as technology improves incentives will quickly no longer be required.

**Keywords:** winery biowaste; combined heat and power; anaerobic digestion; Net Zero

## 1. Introduction

In response to the global climate crisis and the escalating planetary warming resulting from heightened levels of anthropogenic greenhouse gases in the atmosphere, the UK has undertaken the crucial objective of curbing its carbon dioxide (CO$_2$) emissions. This ambitious commitment, known as Net Zero by 2050, entails the reduction of CO$_2$ emissions to levels equivalent to or lower than those recorded in 1990 [1]. The dilemma the UK faces is the inherent conflict between economic growth and the connected escalation of energy consumption and consequent emissions. With a 1% GDP rise generating a 6% surge in consumption [2], the nation finds itself at the juncture of balancing growth with environmental stewardship. A critical mandate arises: emerging industries must align with stringent emission targets while satisfying escalating energy needs.

The emerging British wine industry exemplifies this challenge [3], as advancing technology and rising temperatures make grape cultivation feasible in previously unsuitable regions. Presently, the sector annually consumes nearly 8 million kWh of energy [4,5], a figure projected to increase with industry growth. To fulfil these demands and national Net Zero targets, a renewable net-zero power system could be pivotal for British wineries.

As discussions on renewable energy systems reverberate across wineries worldwide, solar power has commanded significant attention. Its abundance in traditional wine-producing regions and its alignment with the seasonal energy demand profiles of wine-making are primary factors driving this focus [6]. The 2017 LIFE REWIND (Renewable Energy in the Wine Industry) project established a prototype winery in Spain that illustrated the viability of employing photovoltaic (PV) cells to meet energy needs for wastewater treatment and vineyard irrigation [7]. Excess energy was also utilized to generate hydrogen, replacing diesel use. Nonetheless, this project also highlighted the pressing challenge

of energy storage; lead-acid batteries and hydrogen generation proved impractical for long-term solutions.

After the LIFE REWIND initiative, wineries have embraced commercial use of PV cells, yet typically this still covers only a fraction of consumption. By 2021, the González Byass group, spanning Spain, Chile, and Mexico, achieved 20–60% renewable coverage, integrating PV cells alongside biomass, solar thermal, and geothermal sources [8,9]. However, even their most substantial PV installation falls short at 48%. These endeavours illuminate that PV cells alone might inadequately fulfil winery energy demands, particularly in the UK where solar intensity is lower.

Wind power emerges as a compelling alternative, given the UK's leadership in this sector. Accolade Wine in Bristol began to meet 40% of its demand via a single 135 m turbine in 2019, while Lanchester Wines in Durham, currently utilizes 4 smaller turbines to generate 5.5 million kWh/year. Lanchester Wines is now installing wind turbines alongside rooftop PV panels at their new bottling facility which, when operational in 2024, will result in 13% of all British wine being bottled using renewable energy [10,11]. These projects affirm wind energy's technical viability, particularly when used alongside solar, but capital costs and land requirements hinder feasibility for smaller wineries.

Anaerobic digestion for waste treatment and energy production is emerging, motivated by waste valorisation and environmental impact reduction [9,12]. The 2011 AD-Wine project showcased the economic viability of anaerobic digestion for winery wastewater treatment [13]. In 2016 a study was then done into the co-digestion of grape pomace and wastewater and showed that anaerobic digestion could effectively treat both waste streams, producing biogas at a level comparable to other organic sources [14]. However the years since these studies have seen limited commercial adoption due to labour-intensive operation and necessary management of both feedstock and digestate.

Literature indicates that global and British wineries have embraced renewable sources, yet a fully carbon-neutral British winery remains elusive. The need for this is critical, not only because of the expected growth of the British wine making industry but also because: British wineries typically have a larger carbon footprint than those in traditional regions due to higher heating demand [4]; wine produced in the UK typically utilises more machinery, as makers can profit on quantity rather than quality and are not held to tradition and heritage [15]; and many British wineries are located in remote regions and therefore rely on diesel generators for energy production [4]. A locally installed (distributed) renewable power system would address all these issues. Consequently, this paper aims to model, analyse, and assess a system capable of wholly meeting energy requirements for Carvers Hill Estate Winery, situated in southwestern England. The aim is to establish a blueprint for other British wineries.

This paper is organised as follows: Section 2 outlines the case study, followed by a comprehensive top-down and bottom-up analysis of the winery's energy consumption. The top-down method involves using average energy consumption data from different types of British wineries, which is then appropriately scaled for the Carvers Hill Estate Winery. On the other hand, the bottom-up approach assesses the specific activities at Carvers Hill Estate Winery and estimates the energy required for each activity before summing up for each month. Once an annual profile is established, various generation methods are evaluated before an apt renewable power system is proposed. Section 3 then presents technical and economic analyses of this proposed system and non-renewable alternatives. The system originally proposed, a combined solar and biogas power system that consumes biowaste from the winery, is referred to as Option 1. A variation of Option 1, where the anaerobic digester is instead solely fed cattle manure, is denoted Option 1a. Options 2 and 3 are non-renewable alternatives. Option 2 involves a natural gas-powered system; 2a relies on pipeline connections for fuel and 2b uses regular LNG tank deliveries. Option 3 reflects the winery owner's original plan, utilizing biomass for space heating, drawing electricity from the national grid, and relying on diesel for machinery operation. Section 4 discusses the

results of the analysis done in Section 3. The paper concludes by summarizing key findings and identifying avenues for future research in Section 5.

## 2. Methodology

### 2.1. Carvers Hill Estate Winery

Carvers Hill Estate Winery (CHEW) is currently under construction in Shalbourne, Wiltshire. CHEW has 5.55 hectares of 21,000 vines growing Pinot Noir (2.04 ha), Pinot Noir Precoce (1.65 ha), Chardonnay (0.31 ha) and Pinot Meunier (0.55 ha) [16]. CHEW was chosen for the case study as it is a new project and as such represents the current state of British wineries. The winery plans to produce red, white, and sparkling wine with a target of 35,000 bottles a year (265 hL). The winery includes a grain store, retail building, process room, cellar and 4 Dutch cottages. The retail room, process room and cellar are all 185 m$^2$ and the Dutch cottages are 175 m$^2$.

### 2.2. Energy Consumption

Both a top-down and a bottom-up approach were used to estimate energy consumption. The two results were then compared and any differences accounted for to get an ultimate prediction.

Consumption depends heavily on the wine production process. The currently planned process at CHEW was understood from conversations with the winery's stakeholders and analysis of literature [17]. The process is similar for red and white wine, with the major difference being the fermentation temperature (13 °C for white and 25 °C for red). Although sparkling wine requires additional steps, these are not as energy intensive and so the major contributors to the consumption profile for all three varieties are the same: heating, lighting, and transportation equipment. The process for producing red, white, and sparkling wine is shown in Figure 1. To enable a fully electric winery, commercially available electric alternatives to diesel powered tractors, forklifts, mildew sprayer and frost fans were sought and their consumption accounted for in the analysis.

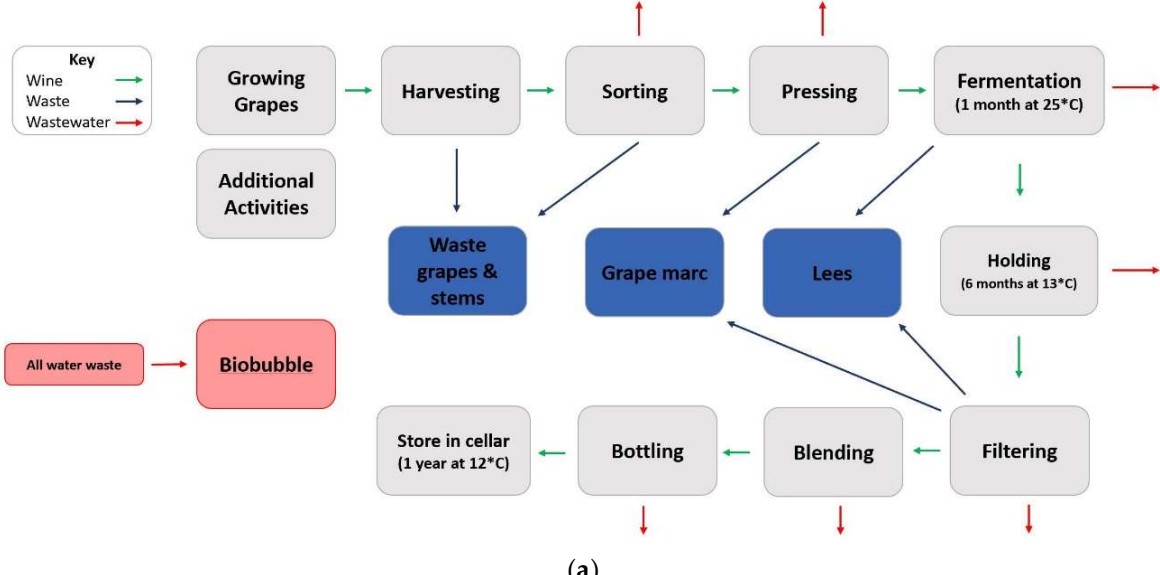

(a)

**Figure 1.** *Cont.*

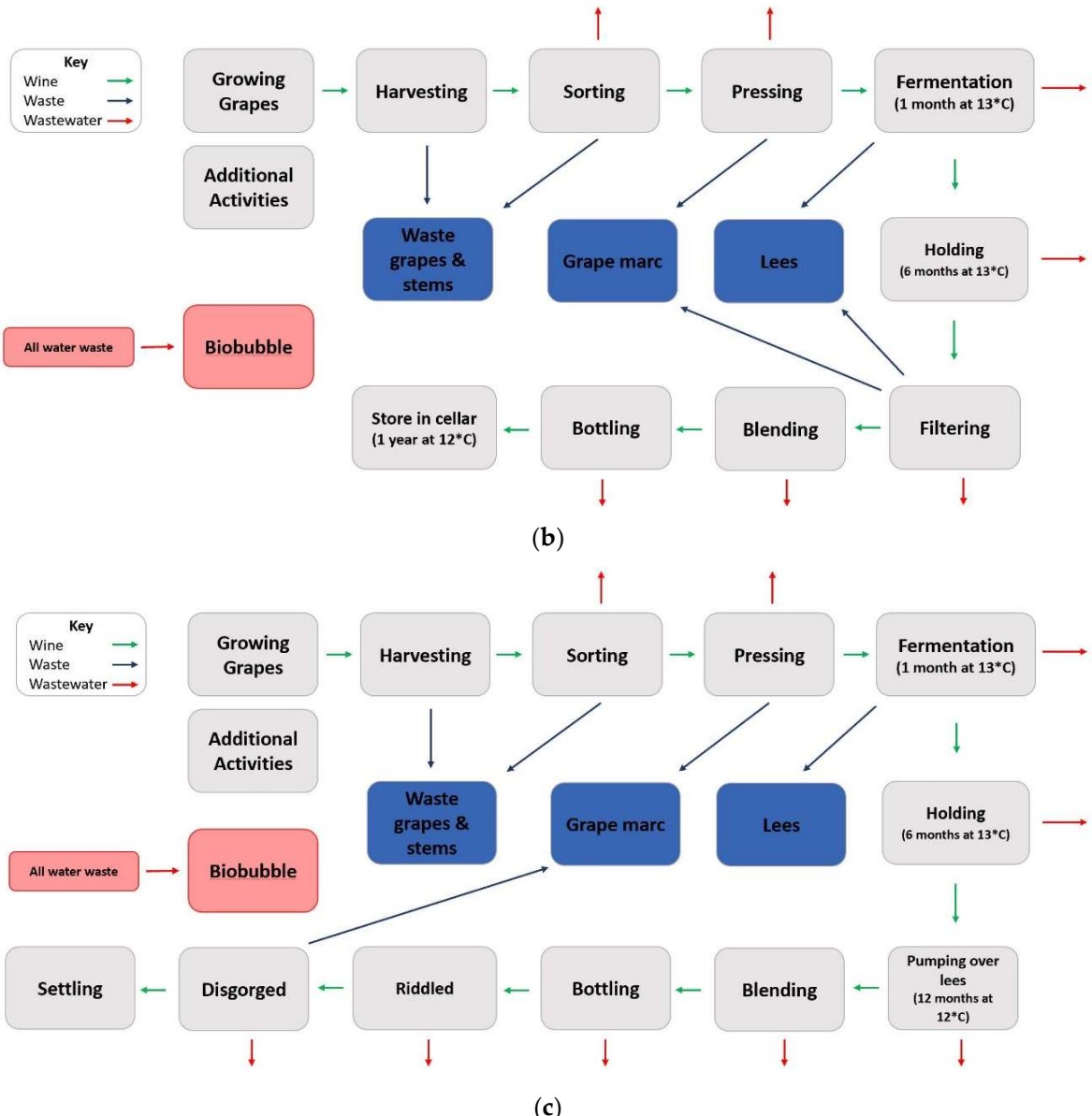

**Figure 1.** The production process for (**a**) Red wine (**b**) White wine (**c**) Sparkling wine at Carvers Hill Estate Winery. (Note: Arrows in the figures are: wine flow in green; waste flow in black; wastewater in red).

### 2.2.1. Top-Down Approach

The results of a 2013 study into the consumption of different sizes and varieties of English and Welsh wineries were used in the top-down analysis [4]. CHEW was modelled first as a medium winery (as its target production is 35,000 bottles/year) and then as a sparkling wine manufacturer. Although CHEW plans to produce a mix of sparkling and still wines, wineries that produce sparkling wine require more energy, so this value was taken. The study covered both electricity and fuel consumption and included ancillary services such as lighting and forklift trucks. However, only the energy consumption after harvest was investigated; energy consumed in growing and harvesting was not accounted for. These were calculated in the following bottom-up approach then incorporated to make a complete profile. The results of the top-down analysis are shown in Figure 2. As can be seen, modelling CHEW as a medium winery resulted in a greater prediction.

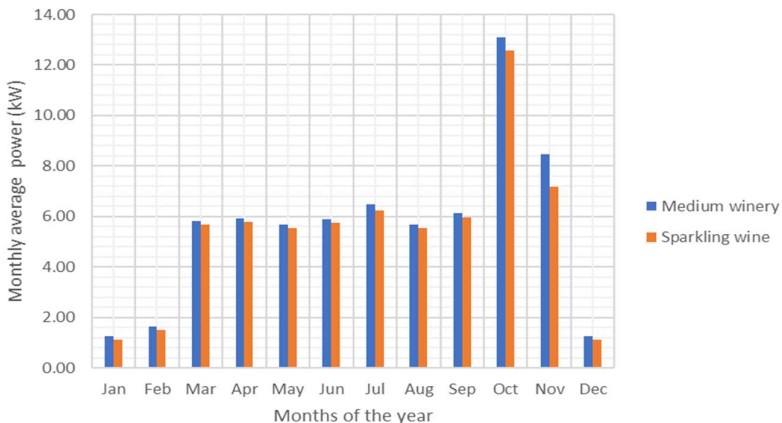

**Figure 2.** Top-down analysis of the consumption of Carvers Hill Estate Winery.

### 2.2.2. Bottom-Up Approach

This approach analysed the energy required for all the pieces of equipment that would be used at each stage of wine production, as specified by the winery's director [17]. Suitable commercially available machinery was considered alongside each device's usage to determine annual consumption from machinery. These results were combined with the lighting, heating and retail consumption discussed below.

### 2.2.3. Heating

Two rooms require temperature regulation: the process room and the retail room. The cellar will have no space heating as the architect has incorporated enough insulation that the room will sustain 12 °C [17]. The cellar will be divided into two rooms, so the inner room remains cold despite the outer door being opened in summer. An underfloor heating system is planned for the retail room and heat pumps were evaluated for both this and the space heating in the process room. The process room will be kept at 14 °C and the underfloor heating system would output 30 °C to keep the retail room a suitable temperature [18]. Both heating systems would not be used when the outside temperature was above 13.5 °C, the average temperature that a British household turns its heating off [19]. The hourly temperature in Marlborough (the nearest town to CHEW) throughout 2021 was used to establish when the heat pumps would be in operation [20]. This data was compared to the UK's average over the last 5 years to confirm typical temperature distribution [21]. The following equations were used to determine the energy required for heating:

$$Q_{out} = A \cdot h \cdot \rho \cdot C_p \cdot (T_{desired} - T_{outside}) \tag{1}$$

$$W_{in} = Q_{out} \cdot SPF \tag{2}$$

where $Q_{out}$ is the heat output, $A$ and $h$ the area and height of the room, $\rho$ and $C_p$ the density and specific heat capacity of air at 20 °C and atmospheric pressure [22,23], and $T_{desired}$ and $T_{outside}$ the desired and outside temperatures. $W_{in}$, the energy input, was calculated using the Seasonal Performance Factor, $SPF$, of the heat pump. The $SPF$ is the average coefficient of performance of a heat pump throughout a year; 2.76 for a domestic heat pump [24]. The hourly energy required for space and underfloor heating was calculated, averaged over the month, and added to the top-down and bottom-up profiles accordingly.

### 2.2.4. Lighting

The average energy consumption for lighting in a commercial building is 75 kWh/m²/year [25]. In total the retail, cellar, and process room equal 1115 m², so the total lighting consumption would be 42 MWh/year.

### 2.2.5. Additional Amenities

The winery will accommodate a range of activities alongside wine production, including wine tasting, weddings, and a shop. The stakeholders plan to host these events in a large retail room equipped with domestic fridge-freezers, an alarm system, Wi-Fi, and other comforts.

The energy consumption of an average Wi-Fi router is 88 kWh/year [26], whilst the average domestic fridge-freezer consumes 404 kWh/year [27]. Two fridge-freezers would be required. A 90 m$^2$ residential building equipped with CCTV and an alarm system consumes 0.68 kWh/day [28]. Therefore, 11 MWh/year is needed for CHEW (built area is 4000 m$^2$). Typically, 16% of the total energy consumption of a winery is due to retail and auxiliary activities [4]. The estimate above amounts to 19% of the bottom-up analysis. The slight increase could be due to the wedding or shop features of the winery, which may not be present in a typical winery.

### 2.2.6. Consumption Results

The bottom-up results, which can be seen in Figure 3, established a consumption profile much larger than the top-down. The main differences were in mixing, lighting, machinery, and retail, where bottom-up exceeded top-down by 3.8 kW, 2.8 kW, 4.5 kW and 5.5 kW respectively. The retail activities at CHEW will be more extensive than at the average English winery [4], so the biggest increase, 5.5 kW, in this area was to be expected. The other differences are likely due to assumptions made during the analysis, where the upper boundary was taken when unsure. Distribution of consumption throughout the year was similar for both results with a large peak in October; slight dip from December to February; and relatively consistent for the rest of the year. The peak in October was to be expected as both the harvest and first fermentation happen in October, so it is when the winery is most active.

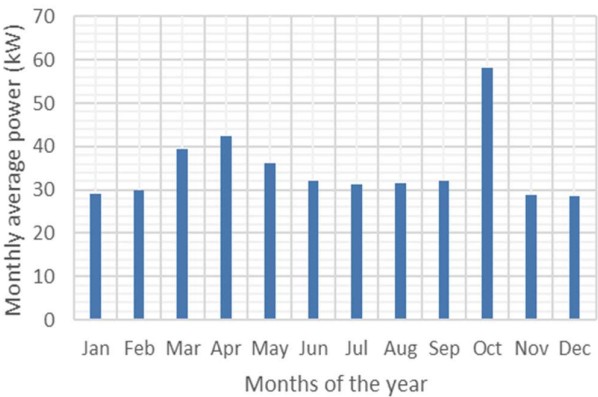

**Figure 3.** Bottom-up analysis of the consumption of Carvers Hill Estate Winery.

The bottom-up results, being larger than that of top-down and more specific to CHEW, were taken for the energy consumption profile the designed renewable power system should supply; over the year 160 MWhe is needed, with 44 MWht heating and 30 MWht cooling. Cooling would later be included in electricity consumption (Section 3) as an alternative to a trigeneration system.

### 2.3. Power Generation

The designed system should cover all areas of the winery, including retail and vineyard. An area of particular interest is frost prevention, as this is fundamental for cooler wine making regions like the UK. Currently the most common means of preventing frost is to light polluting, paraffin bougies [29]. Alternatives are available, including diesel powered fans, however these also increase the carbon footprint of a winery. An electrically driven fan would be required by the renewable system.

Whilst wind power accounts for over half of the UK's national renewable electricity generation [30], it was not deemed suitable for an independent winery because of high upfront costs. Instead, biogas recovered from the wastes of the winery and solar power were considered.

### 2.3.1. Solar

Three of the four Dutch cottages at CHEW have south facing roofs suitable for PV panels. It would therefore be possible to have 156 panels measuring 1.6 m × 1 m. However, by analysing the 2020 and 2021 generation of the 32 such panels installed at Noons Farm, a farm neighbouring CHEW, this would not be enough to meet the demands of CHEW. The panels could be used to offset some of the day-time consumption, as used in the González Byass wineries and LIFE REWIND project [7,8]. Annually, 45 MWhe could be generated, bringing demand down from 160 to 115 MWhe.

### 2.3.2. Biogas

There are four major waste streams throughout the wine making process; the ligno-cellulosic biomass (LCB) waste from growing the grapes, including vine clippings, grape stems, and grape seeds; grape pomace, the solid waste remaining after grape pressing; the lees, the remains from fermentation; and wastewater, primarily from sterilising and cleaning. Table 1 shows the amount of waste expected from CHEW.

**Table 1.** The annual waste at Carvers Hill Estate Winery and expected amount of biogas that could be produced.

| Waste | Literature on Waste Amount | Prediction on Waste for CHEW (Tonnes per Year) | Amount of Biogas Produced ($m^3$/tonne of Waste) | Total Amount of Biogas ($m^3$) |
|---|---|---|---|---|
| Grape stalks | 5 tonnes per hectare per year (7.5% of solid waste) | 27.5 | 25.42 | 700 |
| Grape pomace | 45% of solid waste so 30 tonnes per hectare | 165 | 27.68 | 3110 |
| Grape seeds | 6% of solid waste so 4 tonnes per hectare | 22 | 110.11 | 950 |
| Vine pruning | Linear relationship based on vine planting densities | 10 | 25.42 | 250 |
| Lees | 31,000 tonnes for 1,605,846 tonnes of fruit crushed | 0.29 | 25.72 | 10 |
| Wastewater | 6 litres per litre of wine produced | 90 | 0.17 | 20 |

This waste could be fed into an AD to produce biogas, which could then be burnt in a biogas generator. As shown in Table 1, only 5.02 $m^3$/year of biogas could be produced by the winery waste alone. This would not be enough to power the winery, so other wastes that could be incorporated into the AD were considered.

Potential sources for additional feedstock include food waste, industrial organic residues, and sewage [31]. As CHEW is surrounded by many cattle farms, it was decided that manure would be suitable. Furthermore, a study into the anaerobic digestion of apple pomace noted that manure is one of the most effective additions to the digestion of fruit and vegetable waste, and cattle manure improves yield more than swine or poultry [32]. The study also noted that increasing the proportion of LCB in the feedstock could increase biogas yield. Common sources of LCB are wood, grasses and straw, but grape seeds and vine cuttings are also suitable. The study found the optimum ratio of apple pomace to LCB to cattle manure is 3:2:5. It was assumed that the behaviour of grape pomace would be similar to apple pomace. A 15 kW generator requires 0.00235 $m^3$/s of biogas [33]. The UK-based company, BioQUBE, offers a range of modular ADs. The 2 × 12 m size could generate this amount of biogas [34]. This requires 880 tonnes/year of feedstock. Therefore, in conjunction with the 3:2:5 ratio, 260 tonnes of grape pomace, 180 tonnes of LCB and

440 tonnes of manure would be required throughout the year. As seen in Table 1, CHEW could not supply enough grape pomace and LCB. Additional LCB could be supplied from crop waste of nearby farms, such as Nesbitt Farm which produces both oats and barley, and additional pomace from a local apple press, Kimpton Apple Press [35,36]. Cattle manure could also be sourced from Nesbitt Farm; as a mixed farm with over 200 cows it is estimated that they produce 6600 kg of manure every day [35,37].

Whilst the winery would produce wastewater and wine lees, the amount of biogas produced from these two streams is small compared to the yield of the grape pomace, LCB, and manure mixture (Table 1). Therefore, wastewater and wine lees were not included in the analysis.

Grape pomace will be produced during May, October, and November, when fermenters are emptied or harvest taken in. Energy storage is therefore needed so demand could be met consistently throughout the year. Three possibilities were considered; biomass stored after collection and fed into the AD regularly throughout the year; biomass fed into the AD as it is produced, and the resultant gas stored then fed into the generator when needed; or gas fed directly into the generator during these three peak months, and the electricity stored until required. Whilst the second option would be the most efficient, as the gas could be compressed and stored under pressure [38], the cost of a larger AD would be too great considering it would only be used for three points in the year. For the same reason the third option is dismissed and the first was deemed the most suitable for CHEW.

A combined solar and biogas power system was therefore proposed and is henceforth referred to as Option 1. Option 1 could solely provide electricity or else cogeneration or trigeneration could be employed. Cogeneration requires a combined heat and power (CHP) generator where waste heat from the generator heats the retail and process rooms. Trigeneration uses the waste heat of a CHP generator to power both the heating and cooling of the winery. Schematics detailing these three systems are shown in Figure 4. Alternatively, the winery could be powered by non-renewable sources, either natural gas or the national grid. Such systems will also be analysed to enable comparison and are to be called Options 2 and 3.

Option 2 uses natural gas, either via connection to the UK's gas network (Option 2a) or regular deliveries of liquified natural gas (LNG) tanks (Option 2b). Natural gas could power a CHP generator, and hence used in just power, cogeneration or trigeneration as in the case for Option 1.

Option 3 describes the original intention of the winery owner. Space heating is achieved via a biomass boiler and additional devices, such as the frost fan, run on diesel. Electricity is obtained from the national grid, and an additional diesel generator is onsite in case of power-cuts. Electric fridge-freezers and chillers would be used. This is typical for the British agriculture sector [39].

### 2.4. Modelling and Simulation

ECLIPSE is a simulation software used to assess the technical and economic feasibility of chemical processes by employing the conservation of energy and mass. ECLIPSE is commonly used in studies involving anaerobic digestion, though can model a variety of chemical processes [40]. Anaerobic digestion occurs via four chemical reactions—hydrolysis, acidogenesis, acetogenesis, and methanogenesis—but the overall reaction is given in Equation (3) [41,42]. Similarly, the reactions for combustion of the $CH_4$ in biogas or natural gas is given in Equation (4) [43,44]. The software is user friendly, and analysis is quick once the various parameters have been defined. Therefore, it was deemed a suitable tool for this project. Energy generation from other sources, such as solar, could be estimated separately and added to the results of the simulation.

$$C_6H_{12}O_6 + 8H_2 \rightarrow 5CH_4 + CO_2 + 4H_2O \qquad (3)$$

$$CH_4 + 2(O_2 + 3.76N_2) \rightarrow CO_2 + 2H_2O + 7.52N_2 \qquad (4)$$

Operating ECLIPSE involves four steps. First, a process flow diagram (PFD) must be constructed, detailing the technical performance of components in the system. All the compounds used in the PFD are then defined in the compound database. Next a mass and energy balance can be created, which involves specifying the flow composition, flow rate, and energy transfers across the system. Finally, the software performs a calculation of utilities usage, in which the user selects or adds data from the utility database, and the consumption/production of energy throughout the system is calculated.

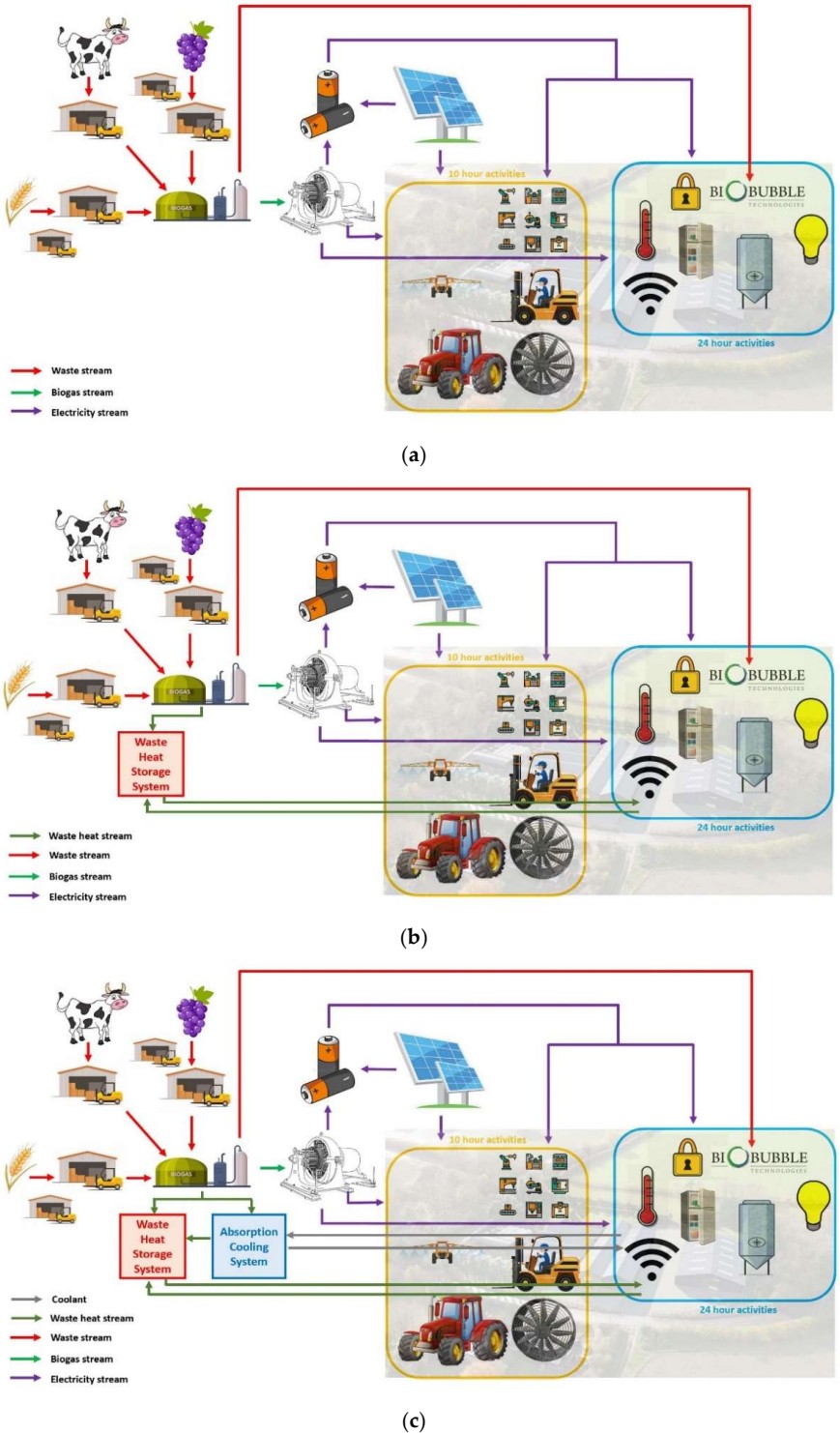

**Figure 4.** Schematics detailing three possible means of providing Carvers Hill Estate Winery's energy demands. (**a**) Electricity demands only (**b**) Cogeneration system and (**c**) Trigeneration system.

## 3. Results

### 3.1. Option 1

The $2 \times 12$ m modular AD sold by BioQube was modelled in ECLIPSE (Figure 5). The AD features a recirculation pipe that feeds 59% of the digestate back through the AD to increase biogas yield [45]. The company states that 2 tonnes/day of food waste can produce 360 m$^3$/day of biogas, with a composition of 60:40 (CH$_4$:CO$_2$) by volume [45]. The ECLIPSE model was verified as biogas produced matched that stated when a feed stock of garbage was used. The winery, however, would use a different feed stream and as such produce a different yield. As mentioned in Section 2, the waste used by the winery would be 30% pomace, 20% LCB and 50% manure. Water would be added to this to reach the optimum solid loading, 10% [46]. The amount added depends on the moisture content of the wastes; this calculation was carried out in MATLAB and used to deduce the overall chemical composition of the feed stream, given in Table 2.

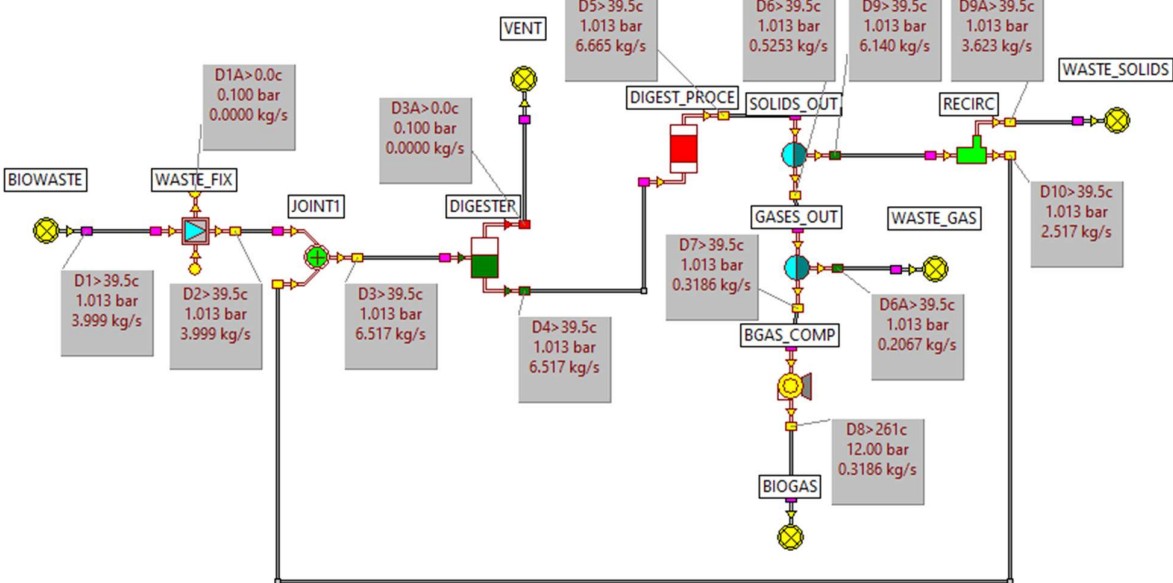

**Figure 5.** ECLIPSE mass-energy balance model of anaerobic digestion.

**Table 2.** Chemical composition of feed streams into anaerobic digester (weight percentages).

|  | Carbon | Hydrogen | Oxygen | Nitrogen | Sulphur | Sodium |
|---|---|---|---|---|---|---|
| Option 1 | 50.01 | 6.17 | 39.39 | 2.75 | 0.98 | 0.70 |
| Option 1a | 42.33 | 6 | 49.12 | 2.55 | - | - |
| Option 2 | 71.90 | 23.86 | 0.57 | 3.67 | 0.00 | 0.00 |

Anaerobic digestion should operate under mesophilic conditions, between 37 °C and 42 °C, to optimise bacteria growth and establish a stable system [47]. Running the mass and energy balance at 39.5 °C found that 0.0032 kg/s of 60:40 (CH$_4$:CO$_2$) biogas would be produced. This is 36% less than that quoted by BioQUBE, but sufficient for supplying a generator [33,45].

A commercially available biogas CHP generator was used to validate the ECLIPSE models shown in Figures 6 and 7 [48]. This generator consumes 0.0029 kg/s of 60:40 biogas and the air/fuel ratio is 6.1. Preliminary results showed the biogas produced by the AD would not be sufficient for a trigeneration system. Instead, a cogeneration system was created, employing electric chillers and fridge/freezers. This would increase electricity consumption to 14 kWe.

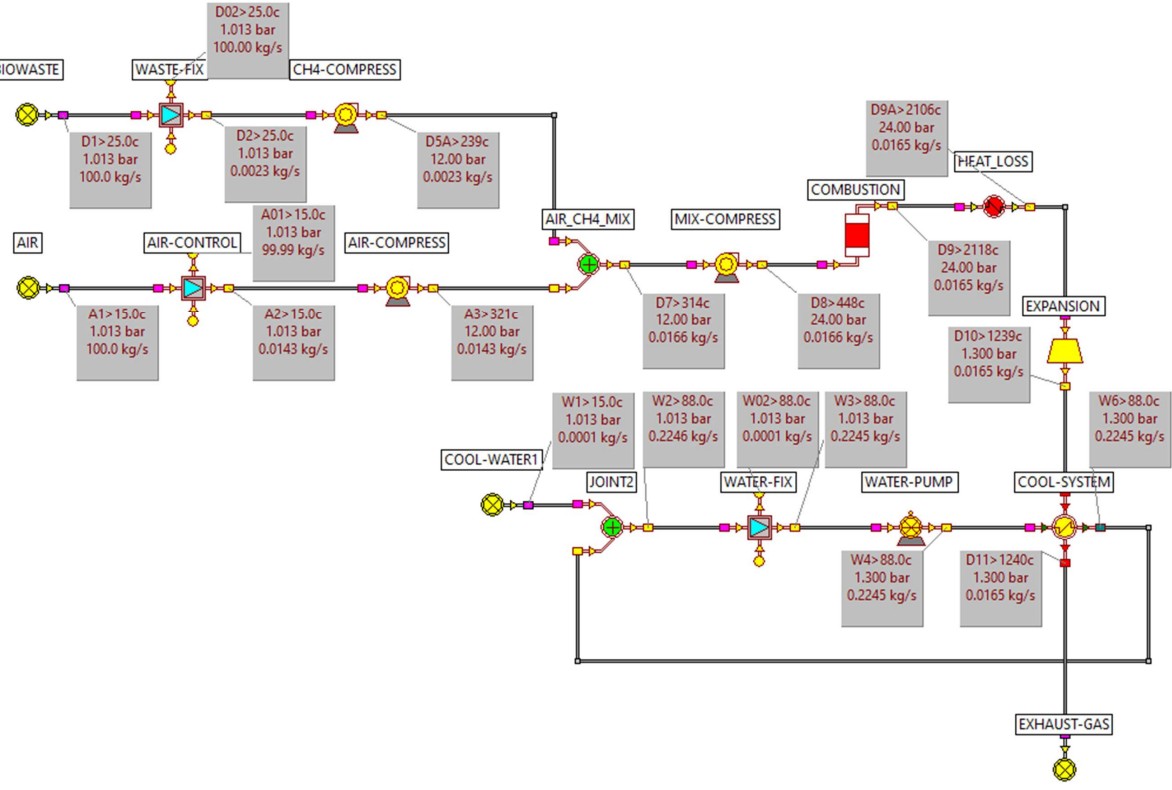

**Figure 6.** ECLIPSE mass-energy balance model of biogas generator (electricity only).

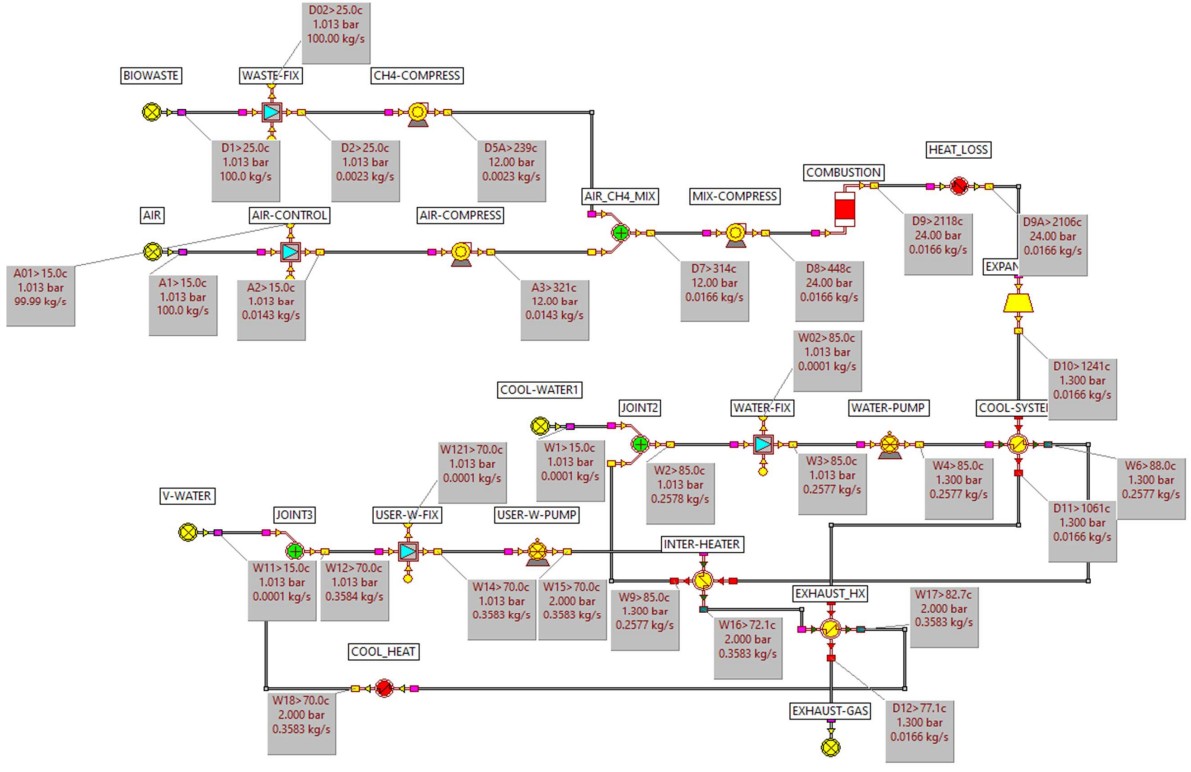

**Figure 7.** ECLIPSE mass-energy balance model of CHP biogas generator.

Figure 6 considers the generator used solely to supply the electricity demand, which it does with 31.8% efficiency. The commercial generator boasts an efficiency of 29.7%, indicating a 7% performance discrepancy [33]. Figure 7 considers cogeneration, with the

generator being used to meet the winery's heating and electricity needs. Electric demand can be met exactly, however thermal energy produced is almost double that required and will be wasted.

### 3.2. Option 1a

It was considered that the cost of storing the winery waste over a year may exceed that of importing biowaste at regular intervals. Furthermore, decomposition would start to reduce the biogas capacity of the stored waste as the year went on [49]. The AD model (Figure 5) was therefore adjusted to a feed stream of just cattle manure, again at 10% total solids. This system will be referred to as Option 1a. It was shown that 0.00270 kg/s of 60:40 biogas would be produced, which is insufficient to power the generator. The modular nature of ADs, however, means production could be increased by 8% by adding another module and increasing the feed stream, making Option 1a technically feasible [50].

### 3.3. Options 2 and 3

The natural gas CHP generator in Option 2 would be similar to that used in Option 1. To evaluate the technical viability of Option 2, the model used in Option 1 was modified to have a natural gas feed stream, the chemical composition of which is shown in Table 2 [51]. The required fuel rate is 0.00101 kg/s and the air/fuel ratio was kept at 6.1. This has an efficiency of 29%, which is closer than that stated by commercial options [48].

Option 3 is assumed to be technically feasible as similar systems are prevalent throughout British wineries and farms [39].

### 3.4. Environmental Impact

The operational $CO_2$ emissions of Options 1, 2 and 3 were assessed and compared. The Current Carbon Intensity (CCI) of the national grid (recorded on 15 March 2023) is 210 $gCO_2$/kWh. This, and the average $CO_2$ emission from burning diesel, predicts that Option 3 would produce 41,100 $kgCO_2$/year. From the ECLIPSE models, Option 2 produced 97,100 $kgCO_2$/year and Option 1 145,100 $kgCO_2$/year. However, the $CO_2$ produced by Option 1 is offset by that consumed by the vines and cattle as the biowaste was being produced. Therefore, the net $CO_2$ emission for Option 1 is zero. These numbers reflect the operational $CO_2$ emissions only and do not account for embodied carbon for the three options. All systems would likely produce net positive emissions in construction [52], nevertheless it is appropriate to conclude that Option 1 will produce significantly less net $CO_2$ over the winery's life.

Another environmental benefit of Option 1 is that it significantly diminishes waste that would otherwise be released into the environment. There are standards in place to ensure waste treatment before release, but it is still desirable to eliminate avoidable waste and double resource productivity, as emphasised by England's Waste Management Plan [53].

### 3.5. Economic Analysis

Economic analysis involved calculating the capital (CapEx) and operational (OpEx) costs of all options and subsequently determining the Levelized Cost of Energy (LCOE). LCOE allows comparison regardless of difference in each option's lifetime and investment, as the cost per unit energy produced over the system's lifetime is determined. LCOE is calculated using Equation (5), where $C_{Cap}$ and $C_{Op}$ are the capital and annual operational costs, $N$ is the lifetime of the system in years, $Q$ the annual energy output, and $d$ the discount rate (which is taken to be 7.5%) [54,55].

$$LCOE = \frac{C_{Cap} + N(C_{Op})}{Q} \cdot \frac{d(1+d)^N}{(1+d)^N - 1} \tag{5}$$

Results of the CapEx, OpEx and LCOE calculations are shown in Table 3. Costs were evaluated first without considering financial incentives and then including carbon tax and

the Renewable Heat Incentive (RHI). The RHI is a scheme by which the government helps meet the cost of installing renewable technologies. The scheme closed for new applicants in March 2022 but was included in analysis to highlight the role of government incentives in making technology economically viable.

**Table 3.** Technical and economic results of study.

| | Option 1 | Option 1a | Option 2a | Option 2b | Option 3 |
|---|---|---|---|---|---|
| **Feedstock** | **Biogas (Winery Waste)** | **Biogas (Cow)** | **Natural Gas (Pipe)** | **Natural Gas (LNG)** | **Grid/Diesel** |
| **Technical results** | | | | | |
| Feedstock input, kg/s | 0.03999 | 0.03999 | - | - | - |
| Bio/Natural gas input, kg/s | 0.00235 | 0.00235 | 0.00101 | 0.00101 | - |
| Total energy input (kW) | 51 | 44 | 49 | 49 | |
| Electrical output, kWe (CHP mode) | 14 | 14 | 15 | 15 | - |
| Heat output, kW$_{thermal}$ (CHP mode) | 22 | 22 | 30 | 30 | - |
| Electrical efficiency, % | 31.82% | 31.82% | 28.7% | 28.7% | - |
| Overall CHP efficiency, % | 71.29% | 81.82% | 91.80% | 91.80% | |
| Heat/electricity ratio (CHP) | 1.57 | 1.57 | 2.00 | 2.00 | - |
| $CO_2$ emissions, kg/year | 145,066 | 145,066 | 97,131 | 97,131 | 41,063 |
| Reduction in $CO_2$ emissions, kg/year | 41,063 | 41,063 | 0 | 0 | 56,068 |
| **Economic results** | | | | | |
| Lifetime | 25 | 25 | 25 | 25 | 25 |
| Feedstock price, £/year | £0.00 | £12,184.62 | £29,925.65 | £121,149.83 | £60,791.26 |
| Capital costs (£) | £674,777.46 | £545,108.60 | £600,524.74 | £200,024.74 | £216,241.52 |
| Operational costs (£) | £64,634.17 | £62,354.38 | £75,516.03 | £138,705.21 | £79,481.81 |
| Total operational cost over lifetime (£) | £1,615,854.28 | £1,558,859.57 | £1,887,900.75 | £3,467,630.25 | £1,987,045.25 |
| LCOE | £1.27 | £1.18 | £1.38 | £1.98 | £1.23 |
| **Economic results (Incentivised)** | | | | | |
| Capital costs (£) | £674,777.46 | £545,108.60 | £600,524.74 | £200,024.74 | £216,241.52 |
| Operational costs (£) | £64,634.17 | £62,354.38 | £83,580.81 | £146,769.99 | £79,618.98 |
| Total operational cost over lifetime (£) | £1,615,854.28 | £1,558,859.57 | £1,887,900.75 | £3,669,249.76 | £1,990,474.43 |
| LCOE (£/kWh) | £1.25 | £1.16 | £1.48 | £2.08 | £1.23 |

Options 2a and 2b represent two different means of accessing natural gas. Option 2a considers establishing a connection to the UK's pipe network which would continuously supply the needs of the winery. As the nearest natural gas pipe is 10 km away from the winery this would involve heavy initial investment. Option 2b considers regular imports of 15 m$^3$ LNG tanks every two months. Whilst this amounts to more over the system's lifetime, it reduces CapEx considerably.

It was decided that Option 1 would utilise the Smart Export Guarantee (SEG) rather than invest in energy storage devices [56]. The sporadic electricity demand is the predominant reason for this; demand nearly doubles during October and to meet this excess energy from the prior two months would have to be stored. This is impractical for two reasons; the dissipation of energy during storage would lead to significant reduction in final capacity and for the remainder of the year less than 10% of the battery would be used, storing excess production overnight for daytime use. These reasons make batteries an inadequate investment and therefore it was decided to buy and sell to the national grid instead. The system could still be considered net zero as overall more energy is sold to the grid than imported. This decision is supported by the energy storage issues found during LIFE REWIND [8]. The results of the LCOE analysis, both incentivised and non-incentivised, are shown for each option in Table 3 along with their technical performance as calculated in this study.

## 4. Discussion

Of the non-renewable systems, Option 3 is the most economical, demonstrated by the LCOE. This is primarily because of the ease of connecting to the national electricity network as opposed to the national gas network.

Although natural gas is advertised as a more sustainable alternative to diesel, due to lower carbon content and higher calorific value, Option 2 produces 80% more $CO_2$ than Option 3 [57]. However, further analysis identifies that only 4% of the $CO_2$ emissions of Option 3 come from diesel use, and the rest are scope 2 emissions from the national grid. As the national grid decarbonises Option 3 would further exceed Option 2 in this area. Nonetheless, both Option 2 and 3 produce net positive $CO_2$, whereas Option 1 (and 1a) can be considered net zero as any $CO_2$ emitted in combustion would have been offset by $CO_2$ consumed during plant and cow growth.

Economically, Option 1 is more apt than Option 2, but Option 3 is still the cheapest. This remains true despite government incentives, such as carbon tax and RHI. Inspection of Table 3 shows that the OpEx for Option 1 is lower than for Option 3, and it is the large CapEx that creates a higher LCOE. Reducing biowaste storage by employing Option 1a reduces the LCOE to £1.16 /kWh, below that of Option 3. This superior performance of 1a over 1 depends heavily on the tax on waste, which is expected to increase in line with England's Waste Management plan [53]. If this tax is greater than the cost of biowaste storage, Option 1 will likely be more economical than Option 1a.

*Sensitivity Analysis*

Changes in technology efficiency, cost, and incentives are highly influential on the comparative performance of Option 1. Option 1 utilises technologies—namely the PV panels and AD—that are rapidly improving in performance and uptake. As such, their efficiency is expected to increase over the coming years and costs decrease. PV panel efficiency is expected to increase by 5% by 2030, and the LCOE decrease by 55% [58,59]. For ADs these figures are 58% and 14% [60,61]. The amenities required to realise Option 1, such as electric vehicles and machinery, are expected to change in the same way. Electric vehicle use is expected to increase from 16% to 50% by 2030 [62], with a cost reduction of 51% [63]. Increased efficiency will also reduce the overall demand of the winery.

If these changes in cost were to take place as expected, the LCOE for Option 1 could be reduced to £0.88 by 2030. Figure 8 shows how independent changes would influence the LCOE of Option 1. The horizontal axis denotes the percentage change in the technology's cost. So far, a discount rate of 7.5% has been used, but this figure could vary between 6–10 [55]. Changes to the RHI are also considered. From the gradients of the graph, it is seen that changes in technology costs and, especially, discount rate have more influence over the LCOE than changes in RHI. Whilst the CapEx/OpEx analysis show incentives like the RHI are useful in making Option 1/1a competitive, it is expected that such incentives would soon not be necessary as technology costs decrease.

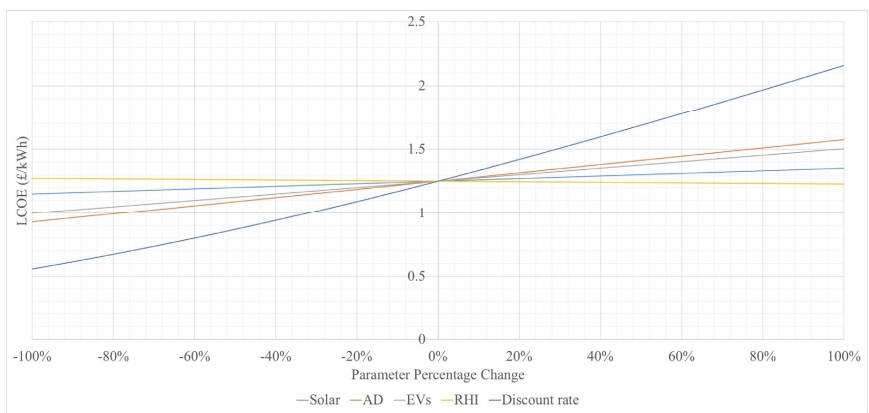

**Figure 8.** Spider diagram showing how changes in technology costs impact LCOE of Option 1.

Cost, efficiency, and tax will likely change adversely for Options 2 and 3 in response to diminishing resources and climate conscious action. Under current policies, the cost of diesel and natural gas are expected to increase by 78% and 61%, and carbon tax by 50% (by 2030) [64,65]. The cost of electricity is expected to remain constant [64]. The Net Zero Emissions by 2050 Scenario, however, anticipates the proportion of renewable, non-pollutant, sources to the national grid will increase from 29% to 60% by 2030 [66]. The grid will also see an increase in small-scale generators utilising the SEG or other flexible schemes [67]. These will therefore become easier to implement or even standard practice, reducing barriers of Option 1.

The results show that Options 1 and 1a are both technically feasible, with 1a the most economical choice of all options at present. Further discussion notes that expected trends will likely reduce these costs and improve technical ability over the next five years. Option 1/1a can bring additional benefits such as odour reduction; a supply of digestate, which is a valuable fertiliser; and certification of sustainably produced wine, which can yield a wider market and higher price. However, there are other barriers to implementing Option 1/1a that are less likely to change with time; namely the space required for PV panels, biowaste storage and AD and the additional time and manpower required for system maintenance. These barriers may improve as the technology develops but are unlikely to compete with the minimal ease and storage of Option 3.

## 5. Conclusions

This study has modelled, analysed, and evaluated a renewable power system for a British winery. The designed system consists of 156 PV panels, $2 \times 12$ m modular anaerobic digester, and a 15 kW CHP biogas generator. By burning biogas and thus avoiding the emissions of a conventional power system, this system saves 41,100 kg$CO_2$/year. The anaerobic digester uses waste from the winery and/or locally-sourced organic matter, alleviating disposal needs and improving sustainability.

The comprehensive LCOE analysis conducted in this study demonstrates the viability of the renewable power system in the face of non-renewable alternatives. Particularly noteworthy is the cost-effectiveness that can be realized through the regular importation of cattle manure over long term storage of winery waste, resulting in a design that is more economically favourable than non-renewable options. The study highlights the pivotal role of governmental support in advancing distributed renewable energy generation. Schemes such as the Smart Export Guarantee are invaluable, as energy storage options are currently unsuitable, but other incentives such as grants, loans and carbon tax help make the LCOE competitive. The anticipated quick reduction in LCOE in response to improvements in technology efficiency and price indicate that this support may soon no longer be needed.

The distributed renewable power system detailed in this paper serves as a template, ready to guide upcoming British wineries in balancing economic development with Net Zero targets. However, the landscape surrounding both the wine industry and climate change mitigation in the UK is evolving. As demonstrated by the sensitivity analysis, it will be important to re-assess the techno-economic performance for different means of distributed power generation in a British winery—both those considered in this study and the new opportunities that will emerge—as this evolution takes place. Research should particularly investigate localised wind generation and energy storage options as the upfront costs of these technologies decrease.

**Author Contributions:** Conceptualization, Y.W.; methodology, S.H.-S. and Y.W.; software, Y.H.; validation, S.H.-S., Y.W. and Y.H.; formal analysis, S.H.-S.; investigation, S.H.-S.; resources, S.H.-S., Y.W. and Y.H.; data curation, S.H.-S.; writing—original draft preparation, S.H.-S.; writing—review and editing, S.H.-S. and Y.W. visualization, S.H.-S. and Y.H.; supervision, Y.W.; project administration, Y.W; funding acquisition, Y.W. All authors have read and agreed to the published version of the manuscript.

**Funding:** This research is partly supported by the Innovate UK funded project "Energy Catalyst Round 9—GAS-SCRIPt (10046865)".

**Institutional Review Board Statement:** Not applicable.

**Informed Consent Statement:** Not applicable.

**Data Availability Statement:** All data are in the paper.

**Acknowledgments:** The authors would like to thank Mr. Griffiths and all those involved with Carvers Hill Estate Winery for their enthusiasm and support in providing information for this study. The authors would like to thank Innovate UK.

**Conflicts of Interest:** The authors declare no conflict of interest.

## Nomenclature

| | |
|---|---|
| A | Area. |
| AD | Anaerobic digester. |
| CapEx | Capital costs. |
| CCI | Current carbon intensity. |
| $CH_4$ | Methane. |
| CHEW | Carvers Hill Estate Winery. |
| $C_6H_{12}O_6$ | Glucose. |
| CHP | Combined heat and power. |
| $CO_2$ | Carbon dioxide. |
| $C_p$ | Specific heat capacity. |
| d | Discount rate. |
| h | Height. |
| $H_2$ | Hydrogen. |
| $H_2O$ | Water. |
| LCB | Lignocellulosic biomass. |
| LCOE | Levelized cost of energy. |
| LNG | Liquified natural gas. |
| N | Lifetime of the system. |
| $N_2$ | Nitrogen. |
| $O_2$ | Oxygen. |
| OpEx | Operational costs. |
| PFD | Process flow diagram. |
| PV | Photovoltaic. |
| Q | Annual energy output. |
| $Q_{out}$ | Heat output. |
| RHI | Renewable heat initiative. |
| SEG | Smart export guarantee. |
| SPF | Seasonal performance factor. |
| $T_{desired}$ | Desired temperature of room. |
| $T_{outside}$ | Temperature outside of room. |
| UK | United Kingdom. |
| $W_{in}$ | Energy input. |
| $\rho$ | Density. |

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
