# Peer review of "Techno-Economic Study of a Distributed Renewable Power System for a British Winery"

_sustainability, doi:10.3390/su151914410_

Round 1

Reviewer 1 Report

 1.     Many abbreviations were used without explaining their meaning.

2.     The technical language/writing of the manuscript needs further improvement and polish. It should be carefully reviewed to enhance the readability.

3.     Moreover, the sentences should be digested and made to the point by removing unnecessary and redundant phrasing. “ The technical writing should be direct and concise”.

4.     The quality of Figures; 4,5, and 6 is very low.  

5.      The existence of many numbers and values in the paragraphs (texts) makes it difficult to compare the results and get the insight of the numbers. A bar graph should be used to visualize many of the number-based paragraphs.

6.      Until the last word in the manuscript, the reviewer believes that this manuscript is a feasibility study or report that some calculation software can generate.

what is the difference between the feasibility study and the work done in this paper?

7.      It is quality is really below a technical/academic article that provides novel findings. The manuscript is really dry without added value!!

- this manuscript is below this journal level!

The technical language/writing of the manuscript needs further improvement and polish. It should be carefully reviewed to enhance the readability.

Author Response

Thank you for your comments. I apologise as I think there may have been some formatting errors with some of the tables/figures in the last document but I believe I have sorted these issues in the file I send through now. I'm afraid figures 5-7 are as good a quality as the ECLIPSE software allows - hopefully there are clear enough to be acceptable? 

Reviewer 2 Report

This reviewer suggests the following points to improve the paper quality:

1.       Try to include the nomenclature of all the symbols used in the work, at the beginning for better readability.

2.       Try to redraft the Introduction section, with background, challenges, literature review, scopes, motivation, contributions, and organization of paper. Highlight the novelties/major contribution of the work prior to organization pf paper in brief(preferably in 3-bulleted points). Also try to expand the literature review including some recent works (of last 3-years) in the similar field.

3.       Try to maintain the work flow of the paper, especially during transition between sections and subsections.

4.       Try to emphasize more on the problem statement and objective function.

5.       Try to quote all the equations in related texts with proper citation (if adopted from published work)

6.       The validation of the proposed method should be provided with comparative analysis. More case studies should be compared to support the claim.

7.       Results should be presented with graphical analysis for better interpretations for different scenarios with source (renewable) and load variations.

8.       Redraft the Conclusion with numerical evidences to support your claim. Also include at least one future scope to it.

9.       Try to redraft the References section with unified formatting as per the journal guidelines.

10.   Proofread the entire manuscript to rectify some existing typos and grammatical errors.

Good

Reviewer 3 Report

The research article "Analysis and Evaluation of a Distributed Renewable Power System for a UK Winery" presents an intriguing design for a renewable energy system tailored specifically for a winery in Wiltshire, England. The paper comprehensively analyses the winery's energy consumption, explores potential generation methods, and proposes a combined design utilizing photovoltaic panels, an anaerobic digester, and a biogas combined heat and power generator. Including specific details, such as using 156 1.6 m x 1 m photovoltaic panels and a modular anaerobic digester, lends credibility to the proposed system's technical feasibility. The research also considers both the technical and economic aspects of the design, using ECLIPSE for technical analysis and comparing the techno-economic performance with traditional power generation methods. One notable highlight of the study is the substantial reduction in the winery's carbon footprint, estimated at 41,100 kgCO2/year. This finding underscores the potential environmental benefits of implementing the proposed system.

Additionally, the research demonstrates that the designed system holds technical viability and competes favorably in the economic landscape due to financial incentives. It is commendable that the authors recognize the impact of technology advancements on system costs. By highlighting that the design's cost primarily fluctuates with technology prices rather than incentives, the study suggests that continued improvements in renewable energy technologies can make incentives less crucial in the future.

This research article presents a well-analyzed and thoughtfully designed distributed renewable power system for a UK winery. The comprehensive examination of the technical and economic aspects, along with the potential environmental benefits, makes this paper a valuable contribution to the field of renewable energy and serves as a possible blueprint for similar applications in the future. However, a few minor revisions are required before it is officially accepted for publication. My detailed comments are as follows;

Abstract. The abstract section is written properly with problem definitions, methodologies, and concise results. Avoid the abbreviations in this section and write a full form of terminologies.

Keywords: Put the keywords in order

Introduction. Usually, at the end of the introduction section, the authors describe the innovations and route of the study. Revise accordingly to put the introduction in the appropriate shape. Further, limit the introduction section and exclude unnecessary repetition. The literature is the latest and most relevant.

Methodology: Methods are appropriate and latest. At the start of the methodology section, add some explanation to describe the importance and significance of these methods for your study.

Data Sources and variables selection: Change the table's caption to variable selection for efficiency estimation. Input-output is too concise and doesn't justify the proper explanation of the variable selection for evaluation.

Figures 5,6 and 7 are not clear; improve the resolution to make it clearer.

Results. Add more citations in this section to explain your results in detail.

Explain the tables and figures near their first mentioned in the text

Carefully check the table number and figure number.

Conclusion. The conclusions are properly written; however, limitations are missing. Add limitations and future research ideas of the study.

Avoid type and grammatical errors throughout the manuscript.

Minor English editing is required.

Author Response

Thank you for your comments. I appologise as I think there may have been some formatting errors with some of the tables/figures in the last document but I believe I have sorted these issues in the file I send through now. I have edited figures 5-7, but unfortunately these are as good a quality as the ECLIPSE software allows - hopefully there are clear enough to be acceptable? 

Round 2

Reviewer 1 Report

Still, the presentation of this paper needs further improvement.  Especially table 3

Addressing the reviewer's comments in a more academic and professional way was expected.

We are all striving to keep the publishing standards of this Journal at its highest level.

Author Response

Dear Editor,

We would like to thank the reviewers for the time and effort they have spent reviewing our paper.

The relevant improvements have been made with the changes tracked in the second version of the revised manuscript.

Yours sincerely

Yaodong Wang (on behalf of all authors)

Dr Yaodong Wang (PhD MSc BSc CEng MIMechE)

Associate Professor, Mechanical Engineering, Department of Engineering

Fellow, Durham Energy Institute

Durham University

Durham, DH1 3LE, UK

Tel:    +44 (0)191 334 2377  

Reviewer #1:

" Still, the presentation of this paper needs further improvement. Especially table 3.

Addressing the reviewer's comments in a more academic and professional way was expected.

We are all striving to keep the publishing standards of this Journal at its highest level."

Response: Thank you for the suggestion. The quality of Table 3 has been improved, as shown below, which has been added into the revised manuscript.

Option 1

Option 1a

Option 2a

Option 2b

Option 3

Feedstock

Biogas (winery waste)

Biogas (cow)

Natural gas (pipe)

Natural gas (LNG)

Grid/Diesel

Technical results

Feedstock input, kg/s

0.03999

0.03999

-

-

-

Bio/Natural gas input, kg/s

0.00235

0.00235

0.00101

0.00101

-

Total energy input (kW)

51

44

49

49

Electrical output, kWe (CHP mode)

14

14

15

15

-

Heat output, kWthermal (CHP mode)

22

22

30

30

-

Electrical efficiency, %

31.82%

31.82%

28.7%

28.7%

-

Overall CHP efficiency, %

71.29%

81.82%

91.80%

91.80%

Heat/electricity ratio (CHP)

1.57

1.57

2.00

2.00

-

CO2 emissions, kg/year

145066

145066

97131

97131

41063

Reduction in CO2 emissions, kg/year

41063

41063

0

0

56068

Economic results

Lifetime

25

25

25

25

25

Feedstock price, £/year

£0.00

£12,184.62

£29,925.65

£121,149.83

£60,791.26

Capital costs (£)

£674,777.46

£545,108.60

£600,524.74

£200,024.74

£216,241.52

Operational costs (£)

£64,634.17

£62,354.38

£75,516.03

£138,705.21

£79,481.81

Total operational cost over lifetime (£)

£1,615,854.28

£1,558,859.57

£1,887,900.75

£3,467,630.25

£1,987,045.25

LCOE

£1.27

£1.18

£1.38

£1.98

£1.23

Economic results (Incentivised)

Capital costs (£)

£674,777.46

£545,108.60

£600,524.74

£200,024.74

£216,241.52

Operational costs (£)

£64,634.17

£62,354.38

£83,580.81

£146,769.99

£79,618.98

Total operational cost over lifetime (£)

£1,615,854.28

£1,558,859.57

£1,887,900.75

£3,669,249.76

£1,990,474.43

LCOE (£/kWh)

£1.25

£1.16

£1.48

£2.08

£1.23
